# Role of Podoplanin (PDPN) in Advancing the Progression and Metastasis of Glioblastoma Multiforme (GBM)

**DOI:** 10.3390/cancers16234051

**Published:** 2024-12-03

**Authors:** Bharti Sharma, George Agriantonis, Zahra Shafaee, Kate Twelker, Navin D. Bhatia, Zachary Kuschner, Monique Arnold, Aubrey Agcon, Jasmine Dave, Juan Mestre, Shalini Arora, Hima Ghanta, Jennifer Whittington

**Affiliations:** 1Department of Surgery, NYC Health and Hospitals, Elmhurst Hospital Center, New York, NY 11373, USA; agriantg@nychhc.org (G.A.); shafaeez1@nychhc.org (Z.S.); bhatian1@nychhc.org (N.D.B.); kushnerz@nychhc.org (Z.K.); bokoa1@nychhc.org (A.A.); davej@nychhc.org (J.D.); mestreju@nychhc.org (J.M.); arorash@nychhc.org (S.A.); ghantah1@nychhc.org (H.G.); harrisj20@nychhc.org (J.W.); 2Department of Surgery, Icahn School of Medicine at the Mount Sinai Hospital, New York, NY 10029, USA; 3Department of Emergency Medicine, Icahn School of Medicine at the Mount Sinai Hospital, New York, NY 10029, USA; monique.arnold@mountsinai.org

**Keywords:** glioblastoma multiforme, podoplanin, venous thromboembolism, platelet aggregation, cancer progression and metastasis

## Abstract

Glioblastoma multiforme (GBM) is a deadly brain tumor, and podoplanin (PDPN) is a transmembrane glycoprotein found in high amounts in GBM. It is regulated by Prox1 and is involved in tumor cell-induced platelet aggregation, tumor metastasis, and lymphatic vessel formation. We do not have enough review literature combining preclinical and clinical evidence to demonstrate PDPN as a standalone prognostic indicator in GBM. This is why we have chosen to present this review.

## 1. Glioblastoma Multiforme

### 1.1. Introduction to Glioblastoma Multiforme (GBM)

Glioblastoma multiforme (GBM) is a frequently occurring malignant primary brain tumor that involves diffusive invasion in the surrounding brain tissues [1]. It is the most aggressive type of brain tumor, categorized as a Grade 4 astrocytic glioma by the World Health Organization (WHO). It typically affects people between 55 and 84 years old and has a high mortality rate [1]. In 2024, it has been projected that there will be 2,001,140 new cases of cancer and 611,720 cancer-related deaths in the United States [2]. In 2023, there were 1,958,310 new cancer cases and 609,820 projected cancer-related deaths in the United States [2]. In 2022, the global incidence of brain and other CNS-associated cancers was 173,690 in males and 148,032 in females [3]. Data from 2014 to 2020 in the United States showed that the prevalence of brain and other nervous system-related cancers varied with age in both males and females across all races and ethnicities. The highest prevalence rate was observed in the 0–19 age group, as shown in Figure 1 [4]. According to the International Agency for Research on Cancer (IARC) in 2022, the highest mortality associated with brain and other CNS cancers in both genders was found in Asia, followed by Europe, Latin America, the Caribbean, North America, Africa, and Oceania [5], as shown in Figure 2. From 2016 to 2020, the average annual incidence rate of all brain and central nervous system (CNS) tumors was 24.83 per 100,000 people, with higher rates in females than males and in non-Hispanic than in Hispanic populations [6]. GBM is more common in males, while meningioma is more common in females [7]. Fifty percent of adult GBM patients die within 10–12 months after diagnosis, and only about 10% survive 24 months after diagnosis [8]. As the survival time increases, patients with GBM may experience cognitive issues, neurological deficits due to radiation-induced tissue damage, communicating hydrocephalus, and, in some cases, cranial neuropathies and polyradiculopathies caused by the spread of cancer cells to the meninges covering the brain and spinal cord [9,10].

British scientific reports in the 1800s were the first to mention gliomas. In 1865, Rudolf Virchow detailed these tumors, recognizing their glial origin [10]. Bramwell noted in 1888 that it is difficult to distinguish between tumor tissue and normal brain tissue without using a microscope because the tumor tissue of GBM is highly infiltrative and does not have a clear boundary [11]. In 1914, the term ‘GBM’ was coined by Mallory [12]. In 1926, Percival Bailey and Harvey Cushing established the basis for the modern classification of gliomas and changed the name from spongioblastoma multiforme to GBM. This term was also favored by Zulch, Russel, and Rubinstein [10]. Hans-Joachim Scherer was the most prolific researcher in glioma research between 1934 and 1941 [13]. In 1940, Scherer discussed how glioma cells migrate away from the primary tumor through the brain. These movement patterns have been referred to as the secondary structures of Scherer. The cells move through the normal brain tissue, gather just below the brain’s surface, surround the neurons and blood vessels, and move through the white matter. This movement may be similar to how cells migrate during the development of the central nervous system [13]. Such behavior causes individual tumor cells to metastasize and reach different parts of the brain. Gliomatosis cerebri is an extreme example of this behavior, where neoplastic cells infiltrate the entire brain diffusely with a minimal or no central focal area of the tumor [14].

The causes of most glioblastomas are unknown, although some established risk factors include inherited genetic syndromes and therapeutic ionizing radiation. However, these factors only account for a small fraction of cases [8]. Risk factors such as body mass index, alcohol consumption, and the use of non-steroidal anti-inflammatory drugs (NSAIDs) have been shown to have a protective effect on GBM risk [15]. There has been no significant evidence linking GBM incidence to diabetes mellitus Type 2 (DM2) or exposure to magnetic fields. Additionally, age, Karnofsky Performance Status (KPS) score, Charlson Comorbidity Index (CCI), the tumor location, and the type of therapy used have been identified as risk factors for complications in GBM patients [16]. Furthermore, Li–Fraumeni syndrome, Neurofibromatosis I, exposure to mutagenic or carcinogenic chemicals, and Turcot syndrome are also considered risk factors for GBM [17]. The dissemination of GBM within the cerebrospinal fluid (CSF) pathways is extremely rare, to the extent that prophylactic irradiation of the spinal cord is not considered justified [10]. The signs of GBM may not be noticeable until the tumor becomes quite large. Typically, the history of the disease is relatively short, often less than a few months, and symptoms can develop suddenly [10]. Podoplanin (PDPN) is often found in high amounts in certain tumors like glioblastomas, and its presence has been linked to a poor prognosis [18]. However, strong clinical evidence for an association between PDPN as an independent prognostic marker and an increased risk of developing GBM is currently lacking. Our objective was to carefully examine numerous studies and deeply analyze the significant role of PDPN in advancing the progression and spread of GBM. Our analysis will assist upcoming researchers in producing robust preclinical and clinical evidence demonstrating PDPN as a distinct prognostic indicator in GBM.

### 1.2. Characteristics and Diagnosis of GBM

Most GBMs develop within brain tissue, with their epicenter located in the white matter. While they typically remain limited to one side of the brain and may affect an entire lobe, some GBMs can be superficial, contacting the protective layers covering the brain but without invading the space between these layers [9]. Primary brain tumors such as meningioma, metastasis, astrocytoma, and oligodendroglioma share some similarities with GBM. GBM of the brainstem, also known as malignant brain stem glioma, is a relatively rare condition that primarily affects children [19]. It accounts for 12–15% of all brain tumors [20]. Within the GBM tumors, there are distinct immunoregulatory macrophages, including spalt-like 1-positive (Sall1+) tumor microglia and Sall1-monocyte-derived macrophages, as well as immunosuppressive T-regulatory cells and dysfunctional T-cell populations, characterized by high levels of cytotoxic T-lymphocyte-associated protein 4 (CTLA-4) and programmed cell death protein 1 [21]. There are three immune response-related subgroups of GBM tumors: negative, humoral, and cellular-like. These are distinguished by their immune cell composition and molecular characteristics [22].

GBM is characterized by features like an irregular mass shape, central necrosis, extensive surrounding edema, and hemorrhage, as well as microscopic variations, such as pseudopalisading necrosis, pleomorphic nuclei and cells, and microvascular proliferation [23]. Necrosis is considered a key diagnostic criterion for upgrading an anaplastic astrocytoma to a GBM at a histopathological level [9]. Considering differential diagnoses, metastasis is less likely in relatively young patients without a history of a primary tumor. Brain abscesses may present with a distinct radiological appearance, characterized by a thin, regular rim of enhancement around a central cavity. Other infections such as toxoplasmosis and cysticercosis may also mimic GBM on imaging [23]. In clinical practice, the biopsy is a common method for obtaining tissue samples for diagnosis, particularly in ~64.1–69% of cases of butterfly GBM. The incidence of multifocal tumors is relatively high (~11.7% to 12.8%), and these patients tend to experience shorter survival compared with those with solitary GBM [24]. Tumor volume and the extent of resection play crucial roles in prognosis, with smaller tumors being more amenable to complete surgical resection. However, not all GBMs are suitable for complete resection, especially in the presence of multiple lesions or larger tumor volumes (for instance, a tumor volume of ≥30 mL) [24]. Therefore, careful consideration of the confounding factors, such as tumor characteristics, is essential when evaluating the extent of resection and its impact on patient survival.

Cellular composition within GBM tumors is highly diverse, reflecting the tumor’s multiforme nature. The tumor cells can exhibit various forms, including fusiform, round, pleomorphic, and astrocytes of different sizes and states of differentiation, which may indicate the emergence of different tumor phenotypes because of stepwise genetic alterations [9]. Molecular and genetic tests have revealed the true diversity of GBM, demonstrating that different genotypes can exhibit the same histomorphology and immunohistochemistry, as well as some aspects of glioma genesis [1]. More than 50% of cases are attributed to the rapid increase in intracranial pressure, which leads to compression of the surrounding brain tissue [10]. The manifestations of GBM can vary, depending on the specific characteristics of the tumor, such as its location, size, and growth rate. Common physical findings associated with GBM include headaches, seizures (experienced by 30–60% of patients), progressive weakness on one side of the body (hemiparesis), and the sudden onset of paralysis on one side with a decreased level of consciousness, which can mimic a stroke in progress and is often caused by bleeding within the tumor [9]. The level of expression of vascular endothelial growth factor (VEGF) is directly related to the occurrence of intratumoral bleeding [25]. Other potential physical manifestations of GBM include sensory disturbances, subtle changes in personality, and visual field defects. In terms of its appearance, GBM is typically diffuse and lacks a distinct outer boundary, exhibiting prominent areas of old and recent hemorrhage (observable as yellowish-brown to red discoloration) and necrotic tissue, which can comprise up to 80% of the total tumor mass [9]. The tumor contains cystic areas mixed with firm tissue. In addition, structures like perineuronal and perivascular “satellitosis” can be found, which are the result of how glioma cells interact with the brain. They are more noticeable in certain parts of the brain, such as the cortex’s subpial zone, the subependymal region, and white matter tracts [9]. When viewed under a microscope, GBM shows specific histopathological features such as various cell and nucleus shapes, abnormal nuclei, high cell division, blood clotting in the vessels, increased small blood vessels, and tissue death [9].

Occasionally, primary CNS lymphoma can appear as a “butterfly-shaped” lesion involving the corpus callosum, while multiple sclerosis lesions, particularly the rare form known as “concentric sclerosis of Balo”, can demonstrate other specific characteristics [25]. The histologic tumor variants such as giant cell glioblastoma, gliosarcoma, and gliomatosis cerebri are generally not associated with a significant impact on the prognosis of the tumor, except gliomatosis cerebri. Gliomatosis cerebri cannot be effectively managed through surgical means and tends to progress more rapidly than other variants [10]. Although the tumor typically extends within and along perivascular spaces, penetration of the vessel lumen appears to be rare, leading to a very low incidence of spread to extra-neural tissues through the bloodstream. The metastatic spread of GBM is infrequent, occurring in less than 5% of cases during the advanced stages of the disease. This phenomenon was nearly unheard of before the introduction of adjuvant therapy [10].

### 1.3. Frequently Mutated Targets/Pathways in GBM

The original belief that GBMs are exclusively derived from glial cells has been challenged by evidence indicating that they may originate from various cell types with properties similar to neural stem cells. These cells are found at different stages of development, ranging from stem cells to neurons to glia. Their characteristics are largely determined by molecular changes in the signaling pathways rather than by their original cell type [26]. Approximately 61% of primary gliomas are located in the brain’s four lobes: frontal (25%), temporal (20%), parietal (13%), and occipital (3%) [27]. Additionally, four GBM subtypes have been identified, each with distinct disease progression and survival outcomes: classical, pro-neural, neural, and mesenchymal [28]. These tumors also exhibit genetic diversity within the tumor cell population [29]. On the basis of their clinical history, GBMs are often classified as primary or secondary [10,30]. Primary GBMs are not associated with a lower malignancy precursor, while secondary GBMs have evolved from a lower-grade astrocytoma [30].

Numerous molecular mutations, distinct sequences of genetic events, and altered pathways associated with uncontrolled cell proliferation and enhanced cell survival contribute to the development of primary and secondary GBMs [18]. TP53 mutations are common in secondary GBMs, occurring in nearly two-thirds of cases, while primary GBMs exhibit p53 mutations in only 25–30% of cases [30]. In total, approximately two-thirds of all GBMs have a wild-type p53 status [10]. For primary GBMs, major factors include EGFR amplification, loss of PTEN or PTEN mutations, loss of chromosome 10q, and INK4A [31]. Secondary GBMs show a longer progression course which involves activation of signal transduction pathways, CDK4 amplification, Rb loss, and loss of PTEN, leading to downstream activation of mTOR, IDH1 mutations, and chromosome 19q loss [28,31]. The mutation rate of the PTEN gene in GBM is ~30.7% [31].

The 2021 World Health Organization (WHO) classification of CNS tumors specifies that glioblastomas are categorized as diffuse astrocytic tumors with isocitrate dehydrogenase (IDH) wild-type (wt). Tumors of astrocytoma with IDH mutant (mut) Grade 2, 3, or 4, are now regarded as distinct entities [32]. The Cancer Genome Atlas (TCGA) analyzed mutational genetic events and identified three core signaling pathways commonly activated in human GBM: the tumor protein p53 pathway, the receptor tyrosine kinase (RTK)/Ras/phosphoinositide 3-kinase signaling pathway, and the retinoblastoma pathway [33,34]. The genomic profiling and the TCGA have been instrumental in providing a thorough understanding of GBM. Through the sequencing of more than 600 genes from over 200 human tumor samples, these efforts have unveiled the complex genetic landscape of GBM [34]. Data from the TCGA project have revealed that genes such as CDKN2A, MDM2, and TP53 are disrupted in approximately 85% of GBM cases [35]. CDKN2A is a gene that is widely expressed in various tissues and cell types. It is located on chromosome 9, band p21.3 [36,37]. The connection of familial GBM37 with mutations in the CDKN2A gene needs further evidence despite the existing association [38]. The methylated-DNA–protein-cysteine methyltransferase (MGMT) promoter is methylated in roughly 50% of newly diagnosed GBM cases as part of the epigenetic events. MGMT produces a DNA repair protein that interferes with the treatment process by eliminating alkyl groups from guanine-rich regions in DNA, which are targets for alkylating agents like temozolomide (TMZ) [39]. The DNA methylation status of this gene might serve as a valuable indicator of the response to chemotherapy and could help to explain why patients with a methylated MGMT gene promoter might experience longer overall survival (OS) [39].

Dysfunction of the p53 pathway contributes to the disruption of the downstream p14ARF pathway, leading to impaired apoptosis and increased genomic instability. Mutated p53 results in the loss of the remaining wild-type alleles, favoring the expression of only abnormal p53 in tumor cells [30]. These findings shed light on the underlying genetic mechanisms at play in GBM and offer potential targets for further research and therapeutic intervention. The presence of mutations in neoplastic and p53-deficient cells is estimated to occur in about 1 in 1000 cells [40]. If this estimate holds for GBMs in vivo, a tumor consisting of 109 cells could potentially harbor up to 106 cells with mutations in a specific gene. Amplification of multiple oncogenes in GBM promotes the dysregulated function of critical tumor suppressor genes. Examples of genetic alterations are overexpressed epidermal growth factor receptor (EGFR) and mutated chromosome 17, or p16/INK4A [41,42]. GBM may have a deletion of parts of chromosomes 1 and 19, known as 1p/19q codeletion. This codeletion is linked to being sensitive to chemotherapy and having a better prognosis in oligodendroglioma [43]. Therefore, it has been suggested that in GBM with an oligodendroglioma component, 1p/19q codeletion may also have a prognostic value [44]. In GBM, the main mechanism of Ras activation is through EGFR, which can be overexpressed and stimulated by external growth factors or be constitutively active in certain mutant variants such as the truncated EGFRIII [45]. Additionally, the binding of various growth factors to other tyrosine kinase receptors represents another crucial upstream event. EGFR, also known as ErbB1/HER1, is a type of RTK that plays a vital role in cell division, migration, adhesion, differentiation, and apoptosis [46].

In glioblastoma (GBM), the progression may be associated with alterations in the Wnt, transforming growth factor beta (TGF-β), VEGF, EGFR, cyclin-dependent kinase 2A (CDKN2A), nuclear factor-κB (NF-κB), and phosphatidylinositol-3-kinase (PI3K)/AKT/mammalian target of rapamycin (mTOR) pathways [47]. VEGF plays a significant role in stimulating endothelial cell growth and regulating normal and pathological blood vessel growth and angiogenesis [47]. CDKN2A acts as a tumor suppressor gene, producing the p16ink4a and p14ARF proteins. The p14ARF protein inhibits murine double minute 2 (MDM2), preventing MDM2-induced degradation of p53 and enhancing p53-dependent transactivation and apoptosis [36]. NF-κB functions as a protein transcription factor and is involved in immunity, inflammation, cancer, and nervous system function [48]. The active form of the NF-κB protein dimer is the heterodimer of p65–p50, which binds to specific κB sites, regulating a wide range of cellular processes [49]. Changes in NF-κB often lead to the development of tumors by promoting tumor growth and invasion, preventing apoptosis, and causing resistance to therapy [50]. The Wnt pathway is involved in development, regeneration, and cellular regulation, impacting the growth, polarity, differentiation, movement, and activity of stem cells [47]. An increase in the activity of the canonical Wnt pathway might result in resistance to chemotherapy and radiation, as well as increased growth, lethality, and invasion in GBM [51]. Normally, TGF-β acts as an inflammatory pathway, encouraging the expression of p21 and other tumor suppressors. However, in cancer cells, TGF-β disrupts the cell cycle and contributes to malignant characteristics [52]. Therefore, the variety of modified signaling pathways points to potential targets that could enhance clinical outcomes in GBM patients.

The pathway PI3K/AKT/mTOR is overactive in GBM. This pathway controls cell dormancy, growth, protein production, cell growth, and lifespan [53]. PI3K becomes active due to various growth factors like the human EGFR family and platelet-derived growth factor receptor (PDGFR) family growth factors. It plays a part in the activation of AKT. AKT is also phosphorylated by the complex mTOR 2 (mTORC2) [54]. Both processes are necessary for full AKT activation. The analysis of the GBM genome revealed significant changes crucial for its development, such as cell cycle checkpoints, apoptosis, pathways related to inflammation, and growth factors [55,56]. In rare pediatric GBM, it is thought that increased expression of YB–1 (Y-Box binding protein 1) reactivates EGFR signaling [57]. The development of glioma involves several stages, including the selective recruitment of mRNAs to polysomes, which enhances protein synthesis, contributes to the development of glioma, and is a key factor in oncogenic Ras and Akt signaling [57]. In addition to the genes discussed, there are many others. The relationship between these genes is illustrated in Figure 3, and a detailed description of these genes can be found in Table 1 of this paper.

One of the reasons for GBM’s resistance to treatment is the complex nature of the tumor. Genetically, it also exhibits various deletions, amplifications, and point mutations that activate signal transduction pathways, as well as disruptions to cell cycle arrest pathways [6]. The standard treatment approach has historically involved extensive surgical removal of the tumor followed by adjuvant radiation therapy (RT), or primary RT for tumors that are not operable. In the past 20 years, both TMZ and a non-invasive device known as the tumor-treating field (TTF; Optune^®^; Novocure GmbH, Root, Switzerland) have shown clinical effectiveness and improved outcomes [18,26,34]. Other promising treatment options include bevacizumab, lomustine, carmustine, PCV (which combines procarbazine, lomustine, and vincristine), and, more recently, the multikinase inhibitor regorafenib, which demonstrated better outcomes than lomustine in a recent Phase 2 trial [58]. Despite advancements in treatment methods, the overall survival rate for GBM patients remains low, with a median survival of only 12–15 months [59]. A proposition suggests that the presence of human cytomegalovirus may impact the cancerous characteristics in glioblastomas [60]. At Karolinska University Hospital, a group of 50 GBM patients was administered valganciclovir as an additional treatment. The survival rate at the 2-year mark was 62%, in contrast to 18% in similar disease stages, surgical resection grades, and baseline treatment contemporary controls [61]. Although these findings seem hopeful, it is still necessary to conduct larger randomized studies to confirm them in the future.

## 2. Podoplanin (PDPN)

PDPN is expressed in both the developing and adult mammalian brain. During the early stages of embryo development, it is widely present on the neuro-epithelium and interacts with the intermediate filament protein nestin within the neural tube [62]. It is widely expressed in various tissues and cell types, for example glomerular podocytes, Type I alveolar cells, osteocytes, mesothelial cells, choroid plexus, glial cells, certain types of neurons such as glutaminergic neurons in the mouse cerebrum [63], cells in the basal layer of human sweat glands, the outer layer of hair follicles, lymphatic endothelial cells (LECs), and different types of fibroblasts [64]. In adult tissues, it plays a crucial role in lymphangiogenesis, platelet production in the bone marrow, and the immune response.

The discovery of PDPN dates back to 1990, when its mRNA was identified in the murine osteoblastic cell line (MC3T3-E1) and ras-transformed cells [65]. Further research has shed light on PDPN’s diverse roles and implications in various biological processes. Two isoforms of PDPN were discovered using Northern blotting. They likely result from alternative splicing, but the biological significance of this discovery is still not fully understood [66,67]. PDPN (also known as Aggrus) is shown to colocalize with ezrin, an ERM-protein (ezrin–radixin–moesin) at the cellular membrane, which promotes the relocalization of ezrin to filopodia-like structures and reduces cell–cell adhesiveness [68]. It is a protein involved in tumor cell-induced platelet aggregation, tumor metastasis, and lymphatic vessel formation [69]. However, the specific way that PDPN causes these processes, including its receptor, has not been fully understood to date.

PDPN has a molecular weight ranging from 36 to 43 kDa. Its coding sequence is present in 201 vertebrate species, spanning from cartilaginous fishes to mammals [69]. It exhibits homologs in humans, mice, rats, dogs, and hamsters and shows a relatively high level of conservation between these species [70]. In humans, it is known as gp36 and T1alpha. Additionally, the widely used synonym for PDPN is D2-40, which is derived from the name of the antibody clone frequently used for its detection in paraffin-embedded tissues [71,72]. Sequences of amino acids (AAs) in PDPN are highly conserved across vertebrates [68]. It was initially identified on lymphatic endothelial cells (LECs) as the E11 antigen, and on the fibroblastic reticular cells (FRCs) of lymphoid organs and thymic epithelial cells as gp38 [70]. The evolutionary history of its coding gene shows that it evolved in jawed vertebrates about 500 million years ago and is absent in jawless fishes and non-vertebrate chordates [69].

PDPN is a transmembrane glycoprotein receptor [67], regulated by Prox1. It is essential for the development of the heart, lungs, spleen, and lymph nodes [73]. It consists of an extracellular domain with approximately 130 AAs, a transmembrane domain with around 25 AAs, and a short intracellular domain with about 10 AAs [68,73]. The extracellular domain is characterized by extensive O-glycosylation, with sialic acid α-2,3 linked to galactose forming the principal component of the protein’s carbohydrate structures [74]. It has a hydrophobic membrane-spanning domain. PDPN does not have recognized functional domains or enzymatic activities. Instead, it interacts with diverse proteins, including C-type lectin-like receptor-2 (CLEC-2), heat shock protein A9 (HSPA9), CD44, galectin 8, chemokine (C-C motif) ligand 21 (CCL21), ezrin, moesin, protein kinase A (PKA), and cyclin-dependent kinase 5 (CDK5), to regulate cell behavior. These ligands and binding partners are crucial in controlling tumor cell migration, cellular annexing, and metastasis [69,70,75,76,77]. The expression of PDPN is stimulated by tumor promoters like TPA, RAS, and Src [65,78,79]. For example, the Src tyrosine kinase uses the focal adhesion adaptor protein Cas/BCAR1 to induce PDPN expression, consequently facilitating tumor cell motility [78]. Src, a nonreceptor protein kinase, promotes non-anchored tumor cell growth and migration, which are vital for invasion and metastasis. Remarkably, Src is not commonly mutated in most cancers, but its activity is associated with various human cancer types, such as brain, colon, breast, pancreas, and skin tumors [80,81].

PDPN expression in multiple types of cancer (e.g., squamous cell carcinoma, glioblastoma, osteosarcoma, bladder cancer, mesothelioma, and seminoma) [76,82] triggers platelet aggregation. It takes a while to get started and relies on the activation of certain proteins. This type of platelet activation is similar to the action of a snake toxin called rhodocytin, and the receptor for this toxin is C-type lectin-like receptor 2 (CLEC-2) [75]. PDPN and a protein called glycoprotein VI (GPVI) are involved in platelet-associated vascular inflammation [75]. PDPN interacts with CLEC-2 in platelets and triggers PDPN-mediated platelet aggregation (PMPA). Chemicals (e.g., platelet-derived growth factor (PDGF), vascular endothelial growth factor (VEGF), serotonin, and coagulation factors) released from the platelets during PMPA help maintain the function of certain blood vessels during immune responses [69]. On platelets, PDPN’s interaction with CLEC-2 is crucial for platelet adhesion, aggregation, maturation, and integrity conservation of the developing cerebral vasculature [62,70]. A study confirmed the association between CLEC-2 and PDPN, which relies on a specific sugar molecule in the structure of PDPN. When CLEC-2 was produced in the laboratory, it was found to inhibit platelet aggregation triggered by tumor cells or lymphatic endothelial cells expressing PDPN [75]. This suggests that CLEC-2 is responsible for the platelet aggregation caused by PDPN on cell surfaces [75]. CLEC-2 is a natural target for PDPN, and PDPN acts as a ligand for CLEC-2 [68,73]. Together, they contribute to PDPN-induced tumor metastasis and related cellular responses [83].

PDPN is made up of eight exons and spans 34.4 kb [67]. The gene coding sequence contains a signal peptide and binding sites for transcription factors such as AP-2, AP-4, C/EBP, and NF-1 in its promoter region [84]. The gene’s 5′-flanking region lacks a TATA box but contains a GATA box of 31 nucleotides upstream of the main mRNA transcription initiation site. Additionally, the gene’s 5′-flanking region has a high guanine–cytosine (GC) content and several potential Sp1 transcription factor sites [84]. In terms of transcriptional regulation, PDPN transcription is enhanced in cells with strongly methylated CpG sites at the promoter region of the gene [84]. Overexpression of the transcription factors Specificity protein 1 (Sp1) and Specificity protein 3 (Sp3) independently increases PDPN transcription in certain cancer cell lines. The sugar modifications in the PDPN ectodomain are typical of mucin O-glycans and involve sialic acid-modified core 1 O-glycans [84]. It induces tyrosine phosphorylation of its cytoplasmic tail by Src family kinases and activates a signaling cascade leading to platelet aggregation/activation. PDPN is also essential for the separation of the lymphatic and blood vasculatures during development [85,86].

The partnership between PDPN and CCL21 in fibroblast-like reticular cells (FRCs) situated in thymic conduits facilitates the development of specialized T-cells [87]. CCL21 serves as a potent chemoattractant in the tumor microenvironment and connects with PDPN-carrying cancer-associated fibroblasts (CAFs), aiding in tumors’ immune evasion [88]. Both the transmembrane domain and cytosolic tail of PDPN, particularly the N-terminal area containing a GXXXG motif (G133IIVG137 in human PDPN), which is crucial in helix–helix oligomerization, have been conserved throughout evolution [69,89]. This specific pattern is extremely important for PDPN’s connection with detergent-resistant membrane (DRM) sections or lipid raft microdomains and holds functional significance. PDPN serves as a substrate for presenilin-1 (PS1)/γ-secretase, which divides its transmembrane domain (between V150 and V151 in human PDPN), releasing the intracellular portion into the cytosol [90]. The proteolysis within the cell membrane of cell surface receptors plays a crucial role in various cellular, physiological, and pathological processes [91], and although the functional importance of the release of the PDPN intracellular domain via γ-secretase is still being studied, it remains significant. Interestingly, both the transmembrane GXXXG pattern and the area near the γ-secretase cleavage point have undergone positive selection throughout evolution [69]. The lack of obvious enzymatic motifs in its structure suggests that PDPN exerts its cellular functions through protein–protein interactions [92]. Despite its wide expression, the precise functions of PDPN in the brain, bones, and tissues like the alveoli, choroid plexuses, and mesothelium, are yet to be fully understood.

## 3. Role of PDPN in GBM

GBM is known for its rapid progression and limited response to treatment. The presence of immunosuppressive macrophages in GBM poses a challenge to the effectiveness of new immunotherapy approaches. Vascular anomalies, including local and peripheral thrombosis, are a common feature of GBM [93]. Studies have indicated that the high expression of PDPN in tumors is associated with an increased risk of venous thromboembolism (VTE) in GBM patients [94]. In a cohort study conducted by Kapteijn et al. (2023), samples of 324 GBM patients were involved in next-generation sequencing of targeted DNA for diagnostic purposes. They identified the mutational status of the genes ATRX, BRAF, CIC, FUBP1, H3F3A, IDH1, IDH2, PIK3CA, PTEN, and TP53, as well as examining the amplification/gain or deletion of BRAF, CDKN2A, EGFR, NOTCH1, and PTEN. The findings proved that the deletion of CDKN2A may play a significant role in glioblastoma-related venous thromboembolism (VTE) [95]. But there are not many studies focusing on PDPN in GBM.

Wu et al. (2024) conducted a comprehensive analysis of Bulk RNA-seq and single-cell RNA-seq of glioma patients from public databases [93]. They performed multiplexed fluorescence immunohistochemistry staining of several markers in glioma tissue microarrays to investigate the impact of a specific marker, PDPN, on macrophages’ immunosuppressive polarization using a co-culture system. The findings revealed an association of PDPN with macrophage M2-like polarization in glioma. Additionally, heterogeneous expression of PDPN at different levels within gliomas was observed. High expression of PDPN was linked to elevated levels of CD68 and certain markers associated with M2-type macrophages [93]. This highlights the role of PDPN-containing extracellular vesicles in regulating the expression of certain genes in macrophages.

Tawil et al. (2023) analyzed the characteristics of cells that express PDPN and their influence on activating the coagulation system and platelets. They validated the nonrandom expression of PDPN among GBM cells and its correlation with cell subpopulations enriched in mesenchymal differentiation signatures, cancer-related inflammation, and coagulation-associated genes [96]. The research also brought to light the inverse relationship between oncogenic drivers and the expression of PDPN in different subgroups of GBM, indicating complex epigenetic mechanisms at play [96]. Moreover, the study proposed that GBM cells expressing PDPN generate a unique type of systemic prothrombotic disturbances in vivo due to extracellular vesicles containing PDPN, underscoring the potential synergy between various coagulants in promoting microthrombosis within the tumor mass [96]. Eisemann et al. (2018) deleted PDPN in patient-derived human GBM cells and found a gene signature associated with high PDPN expression in tumor cells, indicating a poor outcome [97]. The results from the study conducted by Wang et al. (2021) were similar to those of Kapteijn et al. (2023) i.e., high PDPN expression triggers platelet activation and is associated with an increased risk of VTE in patients with cancer [94,98]. Such PDPN-associated mechanisms underlying VTE in patients with GBM highlight the potential implications for cancer-associated venous thrombosis.

Nazari et al. (2018) investigated the presence of the intratumoral IDH1 R132H mutation and PDPN in brain tumor specimens, particularly gliomas, using immunohistochemistry. That study aimed to assess the risk of symptomatic VTE over a 2-year follow-up period. The results indicated a correlation between PDPN expression and IDH1 status in brain tumors. Patients with wild-type IDH1 brain tumors and high PDPN expression showed a significantly increased VTE risk compared with those with mutant IDH1 tumors and no PDPN expression. This shows that brain tumor patients with the IDH1 mutation are at a lower risk of VTE, while the risk in patients with IDH1 wild-type tumors is strongly linked to PDPN expression levels [98,99]. A study involving 197 patients found that 27.4% of them developed VTE, which was linked to a poorer prognosis. However, the study did not find a significant association between the patients’ VTE risk and the Khorana risk score (KRS). This study affirmed that early VTE is a prognostic factor in cancer [99]. Additionally, researchers noted that VTE often occurred within 7 days after surgery, particularly in patients with a lower Karnofsky Performance Scale status and isocitrate dehydrogenase wild-type gliomas expressing PDPN [100]. Their research indicated that most VTEs occur early in the postoperative period and are frequently associated with lower Karnofsky Performance Scale status and isocitrate dehydrogenase wild-type gliomas expressing PDPN [100].

We comprehensively reviewed all relevant preclinical and clinical studies (using clinicaltrial.gov) using the keywords “PDPN” and “GBM”. According to our search, we could not find any study under “podoplanin”, “glioblastoma” and “glioblastoma multiforme” [101]. We recorded clinical studies found under “only glioblastoma but no podoplanin” in Table 2 and those found under “only podoplanin and no glioblastoma or glioblastoma multiforme” in Table 3.

Gi et al. (2022) discovered that prothrombotic factors were expressed within intrathrombus cancer cells, and the thrombus contained erythrocytes, fibrin, platelets, and citrullinated histone H3 and had a certain level of organization when examining the autopsy cases of VTE in cancer (n = 114) and non-cancer patients (n = 66) [102]. Their findings revealed that in 27.5% of deep vein thrombosis cases and 25.9% of pulmonary embolism cases, there was evidence of vascular wall invasion or small cell clusters of cancer cells within the thrombi. Notably, most cancer cells in deep vein thrombi were observed to be invading the vessel wall, whereas the pulmonary embolism cases exhibited cancer cell clusters, suggestive of embolization via blood flow [102]. Furthermore, the study reported that up to 88% of VTE cases showed immunohistochemically positive cancer cells for tissue factors (TF) or PDPN. Additionally, the frequency of TF-positive monocytes/macrophages in the thrombi was higher in cancer-associated VTE cases compared with those without cancer [102]. Histone H3 with citrullination was mainly seen in the initial phases of organizing blood clots. Interestingly, there were no significant differences in the components of blood clots between individuals with cancer-related VTE and those without. The researchers proposed that the presence of cancer cell clusters or invasion of the vascular wall within blood clots may impact the formation of blood clots in VTE associated with cancer. Additionally, the presence of TF and PDPN in cancer cells, as well as monocytes/macrophages, might contribute to blood clotting reactions and the clumping of platelets. The study also suggested that neutrophil extracellular traps might have a role in the early stages of VTE, regardless of the presence of cancer [102].

In one study, brain tumor specimens from 213 patients’ platelet aggregation in response to primary human GBM cells were investigated [103]. The results showed that out of the 151 tumor specimens, 33 had high expression, 47 had medium expression, and 71 had low expression of PDPN [103]. Patients with PDPN-positive tumors had lower peripheral blood platelet counts. The intensity of PDPN staining was associated with higher levels of intravascular platelet aggregates in tumor specimens. Furthermore, high PDPN expression and increased risk of VTE were co-related, independent of age, sex, and tumor type [103]. In 2023, Tawil et al. [96] revealed nonrandom expression of PDPN within GBM cells, with an association with specific cell subpopulations enriched for inflammation- and coagulation-related genes, as well as a mesenchymal differentiation signature. Additionally, the study highlighted a negative association between two oncogenic drivers (EGFR and IDH1 R132H) and the expression of PDPN in distinct GBM subgroups, alongside different epigenetic mechanisms [96]. Furthermore, GBM cells expressing PDPN were found to produce systemic prothrombotic perturbations in vivo, attributed in part to extracellular vesicles carrying PDPN. That study suggested that microthrombosis within the tumor mass may be influenced by the cooperation of major cancer coagulants rather than PDPN alone [96].

Glioma cells release PDPN and/or tissue factors (TFs) within extracellular vesicles, prompting platelet activation and the clotting cascade. Injecting these glioma-derived extracellular vesicles into mice led to increased platelet activation or coagulation markers [96]. The co-expression of PDPN and TF by GBM cells cooperatively contributes to tumor microthrombosis. In conclusion, distinct cellular subsets in GBM drive various aspects of cancer-associated thrombosis and may be targeted for diagnosis and antithrombotic interventions. The researchers suggested that identifying and targeting PDPN-expressing cells or their effects on platelets could help reduce systemic thrombosis associated with GBM.

Cancers that have PDPN expression tend to be more deadly, as they have an increased capacity to generate stem cells, invade the surrounding tissues, and transition from epithelial to mesenchymal cells. This transition contributes to the aggressive advancement of the cancer [104]. In a 2015 study by Grau et al., two human glioma cell lines (U373MG and U87MG) were genetically altered to express PDPN. The effectiveness of this alteration was confirmed through FACS analysis, PCR, and immunocytochemistry. It was observed that the level of PDPN expression varied depending on the grade of the tumors, and all glioblastomas tested positive for PDPN. Moreover, the study indicated that PDPN expression resulted in increased tube formation activity in endothelial cells, regardless of VEGF. These findings suggest that PDPN expression may have a significant impact on tumor progression [105].

Putthisen et al. (2022) identified U373-GSC, a stem-like cell associated with gliomas (GSC). This GSC expressed a specific type of glycan modification involving sialic acid and was common in both GSCs and the parent cell lines [106]. The study demonstrated that the sialic acid modification was highly expressed in GSCs and significantly reduced during the differentiation of GSCs into glioma cells or in the parental cells [106]. This study confirmed that sialic acid modifications (MAL-SGs/alpha-2,3 sialylations) play a role in the survival and maintenance of GSCs, and inhibiting these modifications could lead to the apoptosis of GSCs [106]. In a study, an anti-PDPN monoclonal antibody, humLpMab-23, was humanized, and a defucosylated form, humLpMab-23-f, was produced to enhance its effectiveness in targeting PDPN-overexpressed cells, particularly in GBM. The study demonstrated that humLpMab-23-f was able to induce cellular cytotoxicity and exerted high antitumor activity in mouse xenograft models, suggesting its potential as an antibody therapy for PDPN-positive glioblastomas [104]. Another study investigated the expression of PDPN in glioma tissues and its association with prognostic factors. As a result, PDPN was found to be overexpressed in the majority of glioma tissues compared with normal tissues and was positively correlated with certain prognostic factors such as TERT promoter mutation status and ATRX retention status. Additionally, PDPN knockdown was found to suppress proliferation and reduce protein expression in glioma cells [107].

Astrogliosis is commonly seen in gliomas. Distinguishing between glioma cells and reactive astrocytes [108] can be quite a challenge. Analysis of the genome shows that reactive astrogliosis results in significant diversity among astrocytes, and specific injuries cause changes in particular sets of proteins. Reactive astrogliosis is connected to various abnormal processes in the central nervous system, including gliomas. Malignant gliomas show an increase in PDPN [109]. Research conducted by Kolar et al. (2015) utilizing a syngeneic mouse model of glioma found the substantial expression of PDPN in a specific group of astrocytes expressing glial fibrillary acidic protein within and adjacent to gliomas [109]. This investigation validated the heightened expression of PDPN in astrocytes triggered by glioma growth and brain ischemia, suggesting that PDPN could be a novel cell surface marker for a distinct group of reactive astrocytes near gliomas and non-neoplastic brain lesions. The results also underscore the diversity among reactive astrocytes expressing glial fibrillary acidic protein during gliosis [109]. Previous studies have demonstrated that PDPN, a sialoglycoprotein found in the cell membrane and encoded by the PDPN gene, is upregulated and linked to cellular invasion in astrocytic tumors, although the regulatory mechanisms remain unknown [109].

A study by Cortez et al. (2010) examined miR-29b and miR-125a, which were predicted to regulate PDPN, and revealed that these microRNAs directly target the 3′ untranslated regions of PDPN, inhibiting invasion, apoptosis, and the proliferation of GBM [110]. The study also indicated that miR-29b and miR-125a are downregulated in GBM and PDPN-positive cells, potentially making them a target for GBM therapy [110]. An investigation into the expression of PDPN in astrocytic tumors [111] found that PDPN was present on the surface of anaplastic astrocytoma cells and GBM cells, particularly around necrotic areas and proliferating endothelial cells. However, PDPN expression was not detected in diffuse astrocytomas [111]. Furthermore, they observed markedly higher levels of PDPN mRNA and protein expression in glioblastomas compared with anaplastic astrocytomas, suggesting a potential association between PDPN expression and the malignancy of astrocytic tumors [111]. PDPN is highly expressed in testicular seminoma, suggesting that it may be a sensitive marker for testicular seminomas. Mishima et al. (2006) investigated the expression of PDPN in CNS germ cell tumors (GCTs) by immunohistochemical staining of tumor samples from 62 patients [112]. They found that in 98% of germinomas (including germinomatous components in mixed GCTs), PDPN was diffusely expressed on the surface of germinoma cells [112]. However, non-germinomatous GCTs, including teratomas, embryonal carcinomas, yolk sac tumors, and choriocarcinomas, showed limited or absent PDPN expression [112]. Therefore, PDPN expression may serve as a sensitive immunohistochemical marker for germinoma in CNS GCTs, aiding in diagnosis, monitoring treatment efficacy, and potential antibody-based therapy targeting. A study has shown that lectin from the seeds of Maackia amurensis (MASL) targets PDPN and inhibits transformed cell growth and motility at nanomolar concentrations [113]. This lectin’s biological activity withstands gastrointestinal proteolysis and effectively inhibits cell migration and tumorigenesis [113]. Such studies suggest the potential of using lectins to develop dietary agents targeting specific receptors to combat malignant cells’ growth in GBM.

In recent research conducted by Gharahkhani et al. (2022), they assessed PDPN expression in U87MG cells and observed that the use of the anti-PDPN antibody resulted in decreased cell viability and migration. Additionally, when U87MG cells and platelets were co-treated with and without the anti-PDPN antibody, it was noted that the cells treated with the anti-PDPN antibody displayed a notable reduction in cell platelet aggregation [114]. These results align with existing studies that propose the involvement of PDPN in the aggregation, viability, and invasion of GBM cells, and they suggest that the application of its neutralizing antibody can impede these processes. Research has shown that PDPN identifies a specific subset of aggressive, treatment-resistant glioma cells that contribute to radiation resistance. Modrek et al. (2022) conducted a study on PDPN expression in large glioma cohorts and The Cancer Genome Atlas (TCGA). Their findings suggested that PDPN serves as an independent prognostic marker in glioma patients, and its expression is associated with resistance to radiotherapy and the aggressiveness of tumors. As a result, it has the potential to be a new therapeutic target [115].

Various studies listed in this paper revealed that PDPN is important for GBM’s biology. In GBM, PDPN is connected to alterations in the genetic and epigenetic characteristics of cancer cells, and the specific factors responsible for these alterations are not well understood [96]. Therefore, there is significant interest in understanding how changes in cancer cells’ behavior affect this protein, and blocking PDPN shows promise as a potential therapeutic approach for GBM cancer treatment [114]. Hence, on the basis of the various studies discussed here, it can be said that PDPN is eligible to serve as an independent prognostic marker in GBM.

## 4. Conclusions and Future Perspectives

The analysis in the review extensively examined the emerging link between PDPN and GBM. Its main focus is on the potential of PDPN as an independent prognostic factor in GBM. Our main goal was to comprehensively assess all the relevant preclinical and clinical studies with the highest level of sensitivity to elucidate the role of PDPN in GBM. However, we faced several obstacles, including the varied presentations of GBM. PDPN, as an independent therapeutic target, can introduce complexities in clinical management. As mentioned in the sections above, GBM is typically associated with a median survival rate of 12–15 months following diagnosis. Despite intense approaches like tumor resection, intense radiation, and chemotherapy, the prognosis for GBM patients remains bleak. Existing therapeutic measures only marginally extend patients’ survival and fail to address issues such as therapy resistance and the infiltrative nature of tumor cells into neighboring healthy tissue. Hence, there is an urgent imperative to identify new therapeutic targets and develop more effective treatments to counter therapy resistance and the diffuse spread of tumor cells into surrounding healthy tissue. Additionally, logistical obstacles, such as access to specialized care and healthcare disparities, need to be addressed.

The expression of PDPN in GBM is dependent on the grade of the tumor. PDPN serves as a marker for tumor progression and has been observed to modulate invasion in various neoplasms. However, there is a limited understanding of the function of PDPN in GBM. Overexpression of PDPN has been associated with invasion and metastasis in GBM. Studies have indicated that PDPN expression in glioma cells leads to decreased proliferation, migration, and invasion into a collagen matrix. While functional studies in vivo have been limited to a few publications focusing on other cancer types, there is a lack of specific clinical trials involving PDPN and glioblastoma. Therefore, the evidence regarding PDPN as a therapeutic target for reducing proliferation and metastasis in GBM patients is of low quality. Several research studies have provided indications that PDPN may play a crucial role in driving tumor cell invasion and malignant progression. However, the specific implications of this in high-grade GBM within a preclinical patient-derived xenograft model and clinical studies have yet to be thoroughly examined [96,104]. Some published literature has suggested that PDPN has a tumor-promoting function and has recommended inhibiting the protein as a potential therapeutic approach. Conversely, other studies have proposed that the development and utilization of compounds aiming to functionally inactivate PDPN may not lead to the desired tumor-suppressing effect [97]. Furthermore, there is evidence to suggest that PDPN could serve as a potential prognostic marker in clinical settings and that it is part of a malignant gene signature in GBM, marking highly malicious tumors with a poor prognosis. Nevertheless, it remains to be determined whether targeting PDPN will result in sustained efficacy against GBM or if it might provoke compensating mechanisms that enable the tumor to escape cytotoxic therapy.

The invasive and blood vessel-forming tendencies of GBM, along with the high expression of PDPN, necessitate a thorough investigation into how PDPN affects the interactions between tumor cells and blood vessels, both in laboratory settings and in clinical setups. It is important to recognize that various elements of the tumor’s microenvironment, such as the recruitment of inflammatory cells, the release of cellular chromatin, and the formation of new blood vessels, are controlled by genetic and epigenetic changes in the cancerous cells and their resulting impact on the substances released by the cells. It may be beneficial to explore a wider range of characteristics of cells expressing PDPN in GBM beyond just their PDPN levels, as this could provide additional insights into their involvement in VTE and various biological processes, such as the activation of platelets, the coagulation cascade, and their interactions with inflammatory and vascular cells. A more comprehensive understanding of these cellular interactions, delving beyond individual markers, could potentially offer a new approach to tailoring the management of VTE in GBM. Furthermore, in the realm of molecular biology, there remains a substantial need for further studies to decipher the specific mechanisms through which PDPN operates and to elucidate its therapeutic effects in the treatment of GBM. This exposition underscores the necessity for ongoing research, innovation, and the establishment of standardized protocols to optimize the therapeutic potential of PDPN in the context of GBM.

## Figures and Tables

**Figure 1 cancers-16-04051-f001:**
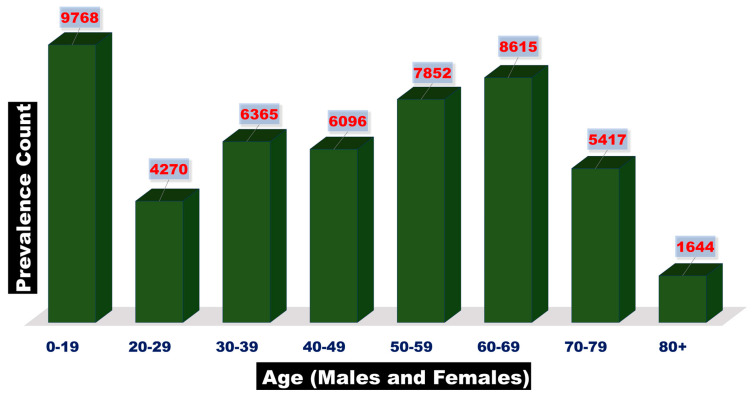
Prevalence counts of the brain- and other nervous system-associated cancers by age in males and females within all races and ethnicities in the United States, 2014–2020 (https://gis.cdc.gov/Cancer/USCS/#/NationalPrevalence/ (accessed on 15 June 2024)).

**Figure 2 cancers-16-04051-f002:**
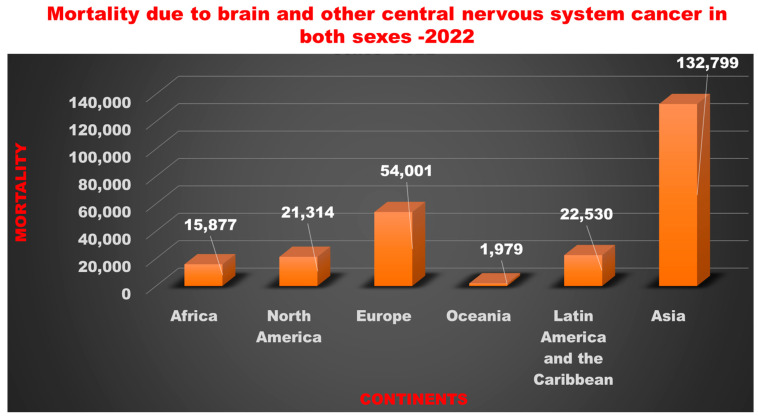
Brain- and central nervous system (CNS) cancer-associated mortality in continents in both sexes in the year 2022. Values were extracted from Globocan, and the authors designed the figure.

**Figure 3 cancers-16-04051-f003:**
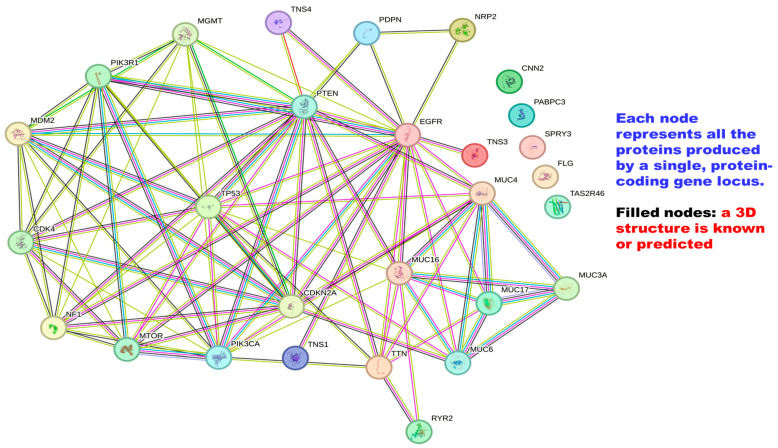
A string figure showing protein-protein associations among various mutated genes in glioblastoma multiforme (GBM). This figure was created with String; available online at https://string-db.org/ (accessed on 27 June 2024).

**Table 1 cancers-16-04051-t001:** Various frequently mutated genes are involved in GBM and their function in unmutated form. The names of all genes were extracted from cBioPortal for cancer genomics, and a full description of these genes was extracted from STRING.

Name of Genes	Full Description
PDPN (podoplanin)	Mediates effects on cell migration and adhesion through its different partners. During development, it plays a role in blood and lymphatic vessel separation by binding CLEC1B, triggering CLEC1B activation in platelets, and leading to platelet activation and/or aggregation.
CDKN2A (cyclin-dependent kinase inhibitor 2A)	Acts as a negative regulator of the proliferation of normal cells by interacting strongly with CDK4 and CDK6. This inhibits their ability to interact with cyclins D and to phosphorylate the retinoblastoma protein. Belongs to the CDKN2 cyclin-dependent kinase inhibitor family.
CDK4 (cyclin-dependent kinase 4)	A Ser/Thr-kinase component of cyclin D-CDK4 (DC) complexes that phosphorylate and inhibit members of the retinoblastoma (RB) protein family including RB1 and regulate the cell cycle during the G (1)/S transition. Phosphorylation of RB1 allows dissociation of the transcription factor E2F from the RB/E2F complexes and the subsequent transcription of E2F target genes which are responsible for the progression through the G (1) phase. Hypophosphorylates RB1 in the early G (1) phase.
MDM2 (mouse double minute 2 homolog)	E3 ubiquitin–protein ligase mediates the ubiquitination of p53/TP53, leading to its degradation by the proteasome. Inhibits p53/TP53- and p73/TP73-mediated cell cycle arrest and apoptosis by binding its transcriptional activation domain. Also acts as a ubiquitin ligase E3 toward itself and ARRB1. Permits the nuclear export of p53/TP53. Promotes proteasome-dependent ubiquitin-independent degradation of retinoblastoma RB1 protein.
MGMT (methylated-DNA–protein-cysteine methyltransferase)	Involved in the cellular defense against the biological effects of O6-methylguanine (O6-MeG) and O4-methylamine (O4-MeT) in DNA. Repairs the methylated nucleobase in DNA by stoichiometrically transferring the methyl group to a cysteine residue in the enzyme.
VEGF (vascular endothelial growth factor)	Growth factor for endothelial cells. It is upregulated in many tumors and its contribution to tumor angiogenesis is well defined.
TNS1,3, 4 (tensin-1,3,4)	Involved in cell migration, fibrillar adhesion formation, cartilage development, and linking signal transduction pathway, and actin remodeling.
MTOR (the mammalian target of rapamycin)	A serine/threonine protein kinase which is a central regulator of cellular metabolism, growth, and survival in response to hormones, growth factors, nutrients, energy, and stress signals. MTOR directly or indirectly regulates the phosphorylation of at least 800 proteins. Functions as part of 2 structurally and functionally distinct signaling complexes, mTORC1 and mTORC2 (mTOR complex 1 and 2). Activated mTORC1 upregulates protein synthesis by phosphorylating key regulators of mRNA translation and ribosome synthesis.
TTN (titin)	A key component in the assembly and functioning of vertebrate striated muscles. By providing connections at the level of individual microfilaments, it contributes to the fine balance of forces between the two halves of the sarcomere. The size and extensibility of the cross-links are the main determinants of the sarcomere extensibility properties of muscle. In non-muscle cells, it seems to play a role in chromosome condensation and chromosome segregation during mitosis.
MUC16 (mucin-16)	Provides a protective, lubricating barrier against particles and infectious agents on mucosal surfaces.
TP53 (Cellular tumor antigen p53)	Acts as a tumor suppressor in many tumor types and induces growth arrest or apoptosis, depending on the physiological circumstances and cell type. Involved in cell cycle regulation as a trans-activator that acts to negatively regulate cell division by controlling a set of genes required for this process.
PTEN (phosphatase and tensin homolog)	Acts as a tumor suppressor and a dual-specificity protein phosphatase, dephosphorylating tyrosine-, serine-, and threonine-phosphorylated proteins.
PABPC3 (polyadenylate-binding protein 3)	Binds the poly(A) tail of mRNA. May be involved in cytoplasmic regulatory processes of mRNA metabolism.
EGFR (epidermal growth factor receptor)	A receptor tyrosine kinase binding ligands of the EGF family and activating several signaling cascades to convert extracellular cues into appropriate cellular responses.
MUC4 (mucin-4 alpha chain)	Plays a role in tumor progression. The ability to promote tumor growth may be mainly due to the repression of apoptosis as opposed to proliferation. Has anti-adhesive properties.
MUC6 (mucin-6)	Provides a mechanism for modulation of the composition of the protective mucus layer related to acid secretion or the presence of bacteria and noxious agents in the lumen. Plays an important role in the cytoprotection of epithelial surfaces and is used as a tumor marker in a variety of cancers. May play a role in epithelial organogenesis
TAS2R46 (taste receptor type 2 member 46)	Plays a role in the perception of bitterness and is gustducin-linked. May play a role in sensing the chemical composition of the gastrointestinal content.
NF1 (neurofibromin truncated)	Stimulates the GTPase activity of Ras. NF1 shows greater affinity for Ras GAP but lower specific activity. May be a regulator of Ras activity.
MUC3A (mucin-3A)	A major glycoprotein component of a variety of mucus gels. Thought to provide a protective, lubricating barrier against particles and infectious agents on mucosal surfaces.
FLG (filaggrin)	Aggregates keratin intermediate filaments and promotes disulfide bond formation among the intermediate filaments during terminal differentiation of the mammalian epidermis.
SPRY3 (protein sprouty homolog 3)	Protein sprouty homolog 3 may function as an antagonist of fibroblast growth factor (FGF) pathways and may negatively modulate respiratory organogenesis. Belongs to the sprouty family.
MUC17 (mucin-17)	Plays a role in maintaining homeostasis on mucosal surfaces.
RYR2 (ryanodine receptor 2)	A calcium channel that mediates the release of Ca^2+^ from the sarcoplasmic reticulum into the cytoplasm and thereby plays a key role in triggering cardiac muscle contraction. Aberrant channel activation can lead to cardiac arrhythmia.
PIK3CA (phosphatidylinositol 4,5-bisphosphate 3-kinase catalytic subunit alpha isoform)	A phosphoinositide-3-kinase (PI3K) that phosphorylates PtdIns (phosphatidylinositol), PtdIns4P (phosphatidylinositol 4-phosphate), and PtdIns(4,5)P2 (phosphatidylinositol 4,5-bisphosphate) to generate phosphatidylinositol 3,4,5-trisphosphate (PIP3). PIP3 plays a key role by recruiting PH domain-containing proteins to the membrane, including AKT1 and PDPK1, activating signaling cascades involved in cell growth, survival, proliferation, motility, and morphology.
PIK3R1 (phosphatidylinositol 3-kinase regulatory subunit alpha)	Binds to activated (phosphorylated) protein-Tyr kinases through its SH2 domain, and acts as an adapter, mediating the association of the p110 catalytic unit with the plasma membrane.
CLO (protein piccolo)	A scaffold protein of the presynaptic cytomatrix at the active zone (CAZ), which is the place in the synapse where neurotransmitter is released (by similarity). After synthesis, it participates in the formation of Golgi-derived membranous organelles termed piccolo–bassoon transport vesicles (PTVs) that are transported along the axons to sites of nascent synaptic contacts.
CNN2 (calponin-2)	A thin filament-associated protein that is implicated in the regulation and modulation of smooth muscle contraction. It is capable of binding to actin, calmodulin, troponin C, and tropomyosin.

**Table 2 cancers-16-04051-t002:** Various clinical studies focusing on glioblastoma [101].

Study Title	Conditions	Interventions	Study Type	NCT Number	Study Status
A study of TAS2940 in participants with locally advanced or metastatic solid tumor cancer	Solid tumor, glioblastoma, non-small cell lung cancer, breast cancer	Drug: TAS2940; drug: TAS2940	Interventional	NCT04982926	Recruiting
A multicenter trial to identify optimal atezolizumab biomarkers in the setting of recurrent glioblastoma. The MOAB trial	Recurrent glioblastoma	Drug: atezolizumab	Interventional	NCT06069726	Recruiting
Efficacy and safety of pembrolizumab (MK-3475) plus lenvatinib (E7080/MK-7902) in previously treated participants with select solid tumors (MK-7902-005/E7080-G000-224/LEAP-005)	Advanced solid tumors, triple negative breast cancer, ovarian cancer, gastric cancer, colorectal cancer, glioblastoma, biliary tract cancers, pancreatic cancer	Biological: pembrolizumab; drug: lenvatinib	Interventional	NCT03797326	Active, not recruiting
Safety and efficacy of transient opening of the blood–brain barrier (BBB) with the SonoCloud-9	Glioblastoma, adult	Device: SonoCloud-9; drug: carboplatin	Interventional	NCT03744026	Completed
Selinexor in treating younger patients with recurrent or refractory solid tumors or high-grade gliomas	Malignant glioma, recurrent brain neoplasm, recurrent childhood central nervous system neoplasm, recurrent childhood glioblastoma, rRecurrent lymphoma, recurrent malignant solid neoplasm, refractory lymphoma, refractory malignant solid neoplasm, refractory primary central nervous system neoplasm, WHO Grade 3 glioma	Other: pharmacological study; drug: selinexor	Interventional	NCT02323880	Active, not recruiting
Oral ONC201 in recurrent GBM, H3 K27M glioma, and midline glioma	Glioblastoma, diffuse midline glioma, H3 K27M glioma, thalamic glioma, infratentorial glioma, basal ganglia glioma	Drug: ONC201	Interventional	NCT02525692	Active, not recruiting
A study to evaluate safety and efficacy of ACT001 and anti-PD-1 in patients with surgically accessible recurrent glioblastoma multiforme	Recurrent glioblastoma multiforme (GBM)	Drug: ACT001; drug: ACT001 + pembrolizumab	Interventional	NCT05053880	Unknown
131I-TLX-101 for treatment of newly diagnosed glioblastoma (IPAX-2)	Neoplastic disease, glioblastoma, glioblastoma multiforme	Drug: 131I-IPA	Interventional	NCT05450744	Recruiting
Efficacy and safety of AP 12009 in patients with recurrent or refractory anaplastic astrocytoma or secondary glioblastoma	Anaplastic astrocytoma, glioblastoma	Drug: trabedersen; drug: temozolomide; device: drug delivery system for administration of AP 12009; procedure: placement of drug delivery system; drug: carmustine; drug: lomustine	Interventional	NCT00761280	Terminated
Vandetanib and sirolimus in patients with recurrent glioblastoma	Glioblastoma	Drug: sirolimus; drug: vandetanib	Interventional	NCT00821080	Completed
PF-00299804 in adult patients with relapsed/recurrent glioblastoma	Glioblastoma, GBM, glioblastoma multiforme	Drug: PF-00299804	Interventional	NCT01112527	Completed
Temozolomide in treating patients with recurrent glioblastoma multiforme or other malignant glioma	Brain and central nervous system tumors	Drug: temozolomide; genetic: protein expression analysis; genetic: reverse transcriptase-polymerase chain reaction; other: diagnostic laboratory biomarker analysis; other: immunoenzyme technique	Interventional	NCT00498927	Completed
Nelfinavir mesylate, radiation therapy, and temozolomide in treating patients with glioblastoma multiforme	Brain and central nervous system tumors	Drug: nelfinavir mesylate; drug: temozolomide; procedure: adjuvant therapy; radiation: radiation therapy	Interventional	NCT00915694	Terminated
Erlotinib and sirolimus in treating patients with recurrent malignant glioma	Brain and central nervous system tumors	Drug: erlotinib + sirolimus	Interventional	NCT00509431	Completed
Open-label trial to explore safety of combining afatinib (BIBW 2992) and radiotherapy with or without temozolomide in newly diagnosed glioblastoma multiform	Glioblastoma	Drug: temozolomide; procedure: radiotherapy; drug: BIBW2992; procedure: radiotherapy; drug: BIBW2992	Interventional	NCT00977431	Completed
Sonobiopsy for noninvasive and sensitive detection of glioblastoma	Glioblastoma, glioblastoma multiforme	Device: sonobiopsy; procedure: research blood; genetic: cancer personalized profiling; device: DefinityÂ^®^	Interventional	NCT05281731	Recruiting
Temozolomide 12 cycles versus 6 cycles of standard first-line treatment in patients with glioblastoma.	Glioblastoma	Drug: temozolomide	Interventional	NCT02209948	Completed
Bevacizumab with or without anti-endoglin monoclonal antibody TRC105 in treating patients with recurrent glioblastoma multiforme	Adult anaplastic astrocytoma, adult anaplastic oligodendroglioma, adult giant cell glioblastoma, adult glioblastoma, adult gliosarcoma, adult mixed glioma, recurrent adult brain neoplasm	Biological: anti-endoglin chimeric monoclonal antibody TRC105; biological: bevacizumab; other: laboratory biomarker analysis; other: pharmacological study; other: quality-of-life assessment	Interventional	NCT01648348	Completed
Bevacizumab w/temozolomide PET & vascular MRI for GBM	Recurrent glioblastoma	Device: MRI-PET; device: TMZ-PET	Interventional	NCT01987830	Completed
Phase 0 analysis of ixazomib (MLN9708) in patients with glioblastoma	Glioblastoma	Drug: ixazomib	Interventional	NCT02630030	Completed
Surgical pembro +/− olaparib w TMZ for rGBM	Glioblastoma, recurrent glioblastoma	Drug: pembrolizumab; drug: olaparib; drug: temozolomide	Interventional	NCT05463848	Recruiting
Detecting malignant brain tumor cells in the bloodstream during surgery to remove the tumor	Astrocytoma, glioblastoma, glioma		Observation	NCT00001148	Completed
Evaluation of GLR2007 for advanced solid tumors	Non-small cell lung cancer, glioblastoma multiforme	Drug: GLR2007	Interventional	NCT04444427	Completed
A study of ribociclib and everolimus following radiation therapy in children with newly diagnosed non-biopsied diffuse pontine gliomas (DIPG) and RB+ biopsied DIPG and high grade gliomas (HGG)	Diffuse intrinsic pontine glioma, malignant glioma of the brain, high-grade glioma, bithalamic high-grade glioma, brainstem glioma, glioblastoma, anaplastic astrocytoma	Drug: ribociclib; drug: everolimus	Interventional	NCT03355794	Completed
Ritonavir and lopinavir in treating patients with progressive or recurrent high-grade glioma	Brain tumor, anaplastic astrocytoma, anaplastic ependymoma, anaplastic oligodendroglioma, brain stem glioma, giant cell glioblastoma, glioblastoma, gliosarcoma, mixed glioma	Drug: ritonavir; drug: lopinavir	Interventional	NCT01095094	Terminated
Safety and tolerability of Fb-PMT in recurrent glioblastoma	Glioma, malignant	Drug: fb-PMT	Interventional	NCT05226494	Recruiting
Bevacizumab and nimustine in patients with recurrent high-grade glioma	Glioblastoma	Drug: bevacizumab; drug: nimustine	Interventional	NCT02698280	COMPLETED
18F-DOPA-PET/MRI scan in imaging elderly patients with newly diagnosed Grade IV malignant glioma or glioblastoma during planning for a short course of proton beam radiation therapy	Glioblastoma, malignant glioma	Procedure: computed tomography; other: fluorodopa F 18; procedure: magnetic resonance imaging; procedure: positron emission tomography; radiation: proton beam radiation therapy; other: quality-of-life assessment; other: questionnaire administration; drug: temozolomide	Interventional	NCT03778294	Completed
Tetra-O-methyl nordihydroguaiaretic acid in treating patients with recurrent high-grade glioma	Brain and central nervous system tumors	Drug: terameprocol; other: pharmacological study	Interventional	NCT00404248	Completed
TN-TC11G (THC + CBD) combination with temozolomide and radiotherapy in patients with newly-diagnosed glioblastoma	Glioblastoma	Drug: TN-TC11G; drug: temozolomide oral product; radiation: radiotherapy	Interventional	NCT03529448	Recruiting
Trial of hypofractionated radiation therapy for glioblastoma	Glioblastoma	Radiation: hypofractionated radiation therapy; radiation: standard radiation therapy	Interventional	NCT02206230	Completed
Ph II bevacizumab + etoposide for Pts w recurrent MG	Glioblastoma, gliosarcoma	Drug: bevacizumab and etoposide	Interventional	NCT00612430	Completed
Monitoring anti-angiogenic therapy in brain tumors by advanced MRI	Glioblastoma	Radiation: avastin; radiation: lomustine; radiation: temozolomide; device: MRI; device: MRS; device: DSC	Observational	NCT02843230	Completed
Surgery for recurrent glioblastoma	Glioblastoma	Procedure: surgery followed by adjuvant second-line therapy; procedure: second-line therapy alone	Interventional	NCT02394626	Recruiting
Intracavitary photodynamic therapy as an adjuvant to resection of glioblastoma or gliosarcoma using IV PhotobacÂ^®^	Glioblastoma multiforme of brain; glioma, sarcomatous	Combination product: photochemotherapy using 3-(1-butyloxy)ethyl-3-diacetyl-bacteriopurpurin-18-n-butylamine methyl ester (PhotobacÂ^®^)	Interventional	NCT05363826	Recruiting
A study of bevacizumab (AvastinÂ^®^) in combination with temozolomide and radiotherapy in participants with newly diagnosed glioblastoma	Glioblastoma	Drug: bevacizumab; drug: temozolomide; radiation: radiation therapy; drug: placebo	Interventional	NCT00943826	Completed
Study evaluating the efficacy and safety of selinexor (KPT-330) in participants with recurrent gliomas	Glioblastoma, glioma	Drug: selinexor	Interventional	NCT01986348	Terminated
Memory-enriched T cells in treating patients with recurrent or refractory Grade III–IV glioma	Glioblastoma, malignant glioma, recurrent glioma, refractory glioma, WHO Grade III glioma	Biological: HER2(EQ)BBÎζ/CD19t + T-cells; other: laboratory biomarker analysis; procedure: leukapheresis	Interventional	NCT03389230	Active, not recruiting
BIBF 1120 for recurrent high-grade gliomas	Glioblastoma, gliosarcoma, anaplastic astrocytoma, anaplastic oligodendroglioma, anaplastic oligoastrocytoma	Drug: BIBF 1120	Interventional	NCT01380782	Completed

**Table 3 cancers-16-04051-t003:** Various clinical trials focusing on podoplanin as a target for cancer studies [101].

Study Title	Conditions	Interventions	NCT Number	Study Statusu
Endoscopic ultrasound plus submucosal injection for early esophageal cancer	Esophageal cancer	Device: submucosal injection needle; device: ordinary endosonography (EUS)	NCT01555801	Unknown
Investigation of contralateral arytenoid sparing IMRT for T1a & T2a larynx cancer & analysis of post-treatment laryngeal function	Laryngeal neoplasms	Radiation: IMRT radiation	NCT02633540	Terminated
Comparison of microwave ablation-assisted enucleation and conventional laparoscopic partial nephrectomy in the treatment of T1a renal cell carcinoma	Renal cell carcinoma	Procedure: laparoscopic microwave ablation-assisted enucleation; procedure: conventional laparoscopic partial nephrectomy	NCT02326558	Completed
Prospective endoscopic follow-up of patients with submucosal and high-risk mucosal esophageal adenocarcinoma	Submucosal esophageal adenocarcinoma, Barrett esophagus, high-risk mucosal esophageal adenocarcinoma	Procedure: endoscopic follow-up	NCT03222635	Recruiting
3D virtual models as an adjunct to preoperative surgical planning	Kidney neoplasms, surgical oncology	Device: 3D-models	NCT03606044	Completed
Outcomes of endoscopically resected high-risk mucosal and low- and high-risk submucosal adenocarcinoma arising in Barrett’s esophagus	Barrett esophagus, adenocarcinoma of the esophagus, submucosal esophageal adenocarcinoma, high-risk mucosal esophageal adenocarcinoma	Procedure: diagnostic endoscopic resection	NCT04818476	Unknown
Percutaneous cryoablation of central kidney tumors with temporary renal artery occlusion	Kidney cancer	NA	NCT05461027	Recruiting
Skipping BCG for T1a urinary bladder tumor	Superficial bladder cancer T1a		NCT04298073	Unknown
Protocol active surveillance small renal masses (SRMs)	Kidney cancer	Other: active surveillance	NCT03804320	Recruiting
Zero ischemia laparoscopic radio frequency ablation assisted enucleation of renal cell carcinoma with T1a stage	Renal cell carcinoma, zero ischemia	Procedure: zero ischemia laparoscopic RFA assisted and ischemia	NCT01838720	Unknown
Using MASL to combat oral cancer	Squamous cell carcinoma of head and neck	Drug: other, placebo	NCT04188665	Recruiting
Multimodal endoscopic image fusion for assessing infiltration in superficial esophageal squamous cell carcinoma	Esophageal neoplasms malignant	Diagnostic test: magnifying endoscopy and endoscopic ultrasonography	NCT06412419	Not yet Recruiting
Transperitoneal vs. retroperitoneal laparoscopic or robotic partial nephrectomy	Renal cell carcinoma	Procedure: laparoscopic or robotic partial nephrectomy	NCT02849119	Unknown
DAHANCA 27 transoral laser microsurgery for T1a glottic cancer	Head and neck squamous cell carcinoma, glottic T1	Procedure: transoral laser microsurgery	NCT05289336	Unknown
Patient reported outcome after nephron sparing treatment of small renal tumours	Renal cell carcinoma	Procedure: CT-guided cryoablation and partial nephrectomy	NCT04040530	Active and not recruiting
Comparison between partial nephrectomy and ablation for renal tumor	Renal cell carcinoma	Procedure: partial nephrectomy and microwave ablation	NCT03094949	Unknown
Air dissection in percutaneous radiofrequency ablation of T1a renal cell carcinoma	Unrecognized condition	Procedure: RFA	NCT03395379	Completed
Surgery versus stereotactic body radiation therapy for stage up to IA2 (T1a or T1b) non-small cell lung cancer	Non-small cell lung cancer	Procedure: anatomical segmentectomy, lobectomy or bilobectomy; Radiation: stereotactic body radiation therapy	NCT03431415	Withdrawn
Voice S: voice quality after transoral CO_2_-laser surgery versus single vocal cord irradiation for larynx cancer	Glottis tumor, larynx cancer	Radiation: single vocal cord irradiation (SVCI); Procedure: Arm A: transoral CO_2_-laser microsurgical cordectomy (TLM)	NCT04057209	Recruiting
ESD for colorectal LSL using a selective strategy—a prospective cohort study	Colorectal neoplasm	Procedure: endoscopic submucosal dissection and endoscopic mucosal resection	NCT04008407	Recruiting
Vascular targeted photodynamic therapy T1a renal tumors	Renal cancer	Drug: light-activated WST11	NCT01573156	Terminated
Radiation therapy plus amifostine in treating patients with primary prostate cancer	Prostate cancer	Drug: amifostine trihydrate; Radiation: radiation therapy	NCT00003307	Completed
Submucosal saline injection followed by endoscopic ultrasound	Esophageal cancer	Procedure: submucosal saline injection, blue laser imaging and magnifying endoscopy	NCT06022978	Recruiting
Oncolytic adenovirus-mediated gene therapy for lung cancer	Non-small cell lung cancer, Stage I	Biological: Ad5-yCD/mutTKSR39rep-ADP adenovirus	NCT03029871	Withdrawn
Robot-assisted laparoscopic radio frequency ablation assisted enucleation of renal cell carcinoma with T1a stage	Renal cell carcinoma	Procedure: zero ischemia robot-assisted laparoscopic RFA-assisted and zero ischemia robot-assisted laparoscopic partial nephrectomy	NCT02924597	Unknown
A study of LY3022855 in combination with durvalumab or tremelimumab in participants with advanced solid tumors	Solid tumor	Drug: LY3022855; Drug: durvalumab; Drug: tremelimumab	NCT02718911	Completed
Measuring the impact of MammaPrint on adjuvant and neoadjuvant treatment in breast cancer patients: A prospective registry	Breast cancer	NA	NCT02670577	Completed

## Data Availability

Information for this review was collected from various sources including PubMed, Globocan, World Cancer Research Fund International, National Brain Tumor Society, String, US Centers For Disease Control and Prevention, Google Scholar, Scopus, Web of Science, and ClinicalTrials.gov.

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
