# Peer review of "Role of Podoplanin (PDPN) in Advancing the Progression and Metastasis of Glioblastoma Multiforme (GBM)"

_cancers, 2024, doi:10.3390/cancers16234051_

Round 1
Reviewer 1 Report
Comments and Suggestions for Authors
The manuscript entitled "Role of podoplanin (PDPN) in advancing the progression and 2 metastasis of glioblastoma multiforme (GBM)" by Bharti Sharma and colleagues is a broad purpose narrative review of glioblastoma and podoplanine. The initial part of the review (up to row 352) contains a generic exposition piling up well known facts about GBM and various glioma in humans with little if any mention to PDPN. The length of this section of the review, in my view is too extensive for a review whose title is focusing on the role of PDPN in gliomas. The second part of the review is more appropriately focused on PDPN but, it still contains many repetitions. Furthermore, the authors do not clearly discuss if PDPN is important for GBM biology because it modulates the action of some of the reactive astrocytes present in and around the tumor or if it role is mostly linked to the expression of the gene in a subpopulation of GBM cells. A more compact and incisive presentation of the literature would greatly improve the present version of the review.
Minor point:
p. 22 row 625 " The glycoprotein PDPN (PDPN)". the abbreviation PDPN was defined long before you can use it without further indications.
Author Response
Greetings,
Thanks for your comments
Reviewer 1:
- The initial part of the review (up to row 352) contains a generic exposition piling up well known facts about GBM and various glioma in humans with little if any mention to PDPN.
Response: We have designed an outline for our paper in the following sequence:
PART 1: A Glioblastoma multiforme (GBM)
- Introduction on Glioblastoma multiforme (GBM)
- Characteristics and diagnosis of GBM
- Frequently Mutated Targets/Pathways in GBM
PART 2: B. Podoplanin
- Introduction on PDPN
- Role of PDPN in GBM
As mentioned by reviewer 1, we have tried to include all possible genetic mutation as part of explaining GBM in part 1. We have re-written entire section for an ease of understanding and removed all duplication.
- The length of this section of the review, in my view is too extensive for a review whose title is focusing on the role of PDPN in gliomas. The second part of the review is more appropriately focused on PDPN but, it still contains many repetitions.
Response: a) As this paper focuses on PDPN in GBM, without describing GBM and role of PDPN in GBM cannot be explained. We have tried to emphasize on GBM’s characterstics, diagnosis and mutated pathways
- b) We have tried to incorporate findings from most of the published research studies (clinical and preclinical). There is not sufficient literature focusing on PDPN in GBM.
- c) We have re-written second part of this paper and now there are no repetitions
- d) Part 2 of our paper titled as role of PDPN in GBM is full of such details. Few examples are listed below:
- Lines # 571-576: Another study investigated the expression of PDPN in glioma tissues and its association with prognostic factors. As a result, PDPN was found to be overexpressed in the majority of glioma tissues compared to normal tissues and was positively correlated with certain prognostic factors such as TERT promoter mutation status, and ATRX retention status. Additionally, PDPN knockdown was found to suppress proliferation and reduce protein expression in glioma cells [107].
- Lines # 444-454: Wu et al. (2024), conducted a comprehensive analysis of Bulk RNA‐seq and single‐cell RNA‐seq of glioma patients from public databases [93]. They performed multiplexed fluorescence immunohistochemistry staining of several markers in glioma tissue microarray to investigate the impact of a specific marker, PDPN, on macrophage immunosuppressive polarization using a co‐culture system. The findings revealed an association of PDPN with macrophage M2‐like polarization in glioma. Additionally, heterogeneous expression of PDPN at different levels within gliomas was observed. High expression of PDPN was linked to elevated levels of CD68 and certain markers associated with M2‐type macro-phages [93]. This highlights the role of PDPN-containing extracellular vesicles in regulating the expression of certain genes in macrophages.
- Lines # 473-481: Nazari et al (2018), investigated the presence of intratumoral IDH1 R132H mutation and PDPN in brain tumor specimens, particularly gliomas, using immunohistochemistry. This study aimed to assess the risk of symptomatic VTE over a 2-year follow-up period. The results indicated a correlation between PDPN expression and IDH1 status in brain tumors. Patients with wild-type IDH1 brain tumors and high PDPN expression showed a significantly increased VTE risk compared to those with mutant IDH1 tumors and no PDPN ex-pression. This shows that brain tumor patients with IDH1 mutation are at a lower risk of VTE, while the risk in patients with IDH1 wild-type tumors is strongly linked to PDPN expression levels [98,99].
- Lines # 481-484: A study involving 197 patients found that 27.4% of them developed VTE, which was linked to a poorer prognosis. However, the study did not find a significant association between the patient's VTE risk and the Khorana risk score (KRS). This study affirmed that early VTE is a prognostic factor in cancer [99].
- Lines # 484-490: Additionally, researchers noted that VTE often occurred within 7 days after surgery, particularly in patients with lower Karnofsky Performance Scale status and isocitrate dehydrogenase-wildtype gliomas expressing PDPN [100]. Their research indicated that most VTEs occur early in the post-operative period and are frequently associated with lower Karnofsky Performance Scale status and isocitrate dehydrogenase-wildtype gliomas expressing PDPN [100].
- Furthermore, the authors do not clearly discuss if PDPN is important for GBM biology because it modulates the action of some of the reactive astrocytes present in and around the tumor or if it role is mostly linked to the expression of the gene in a subpopulation of GBM cells.
Response: Part 2 of our paper titled as role of PDPN in GBM is full of such details. Few examples are listed below. Various studies listed in this paper reveals that PDPN is important for GBM biology. In GBM, PDPN is connected to alterations in the genetic and epigenetic characteristics of cancer cells, and the specific factors responsible for these alterations are not well understood [96]. Therefore, there is significant interest in understanding how changes in cancer cell behavior affect this protein, and blocking PDPN shows promise as a potential therapeutic approach for GBM cancer treatment [114]. Hence, based on various discussed studies, it can be said that PDPN is eligible to serve as an independent prognostic marker in GBM. Since the focus of this paper is not Astrocytes but GBM, so we have only included limited information on it.
- Lines # 583-593: Research conducted by Kolar et al (2015) utilizing a syngeneic mouse model of glioma found a substantial expression of PDPN in a specific group of astrocytes expressing glial fibrillary acidic protein within and adjacent to gliomas [109]. This investigation validated the heightened expression of PDPN in astrocytes triggered by glioma growth, and brain ischemia, suggesting that PDPN could be a novel cell surface marker for a distinct group of reactive astrocytes near gliomas and nonneoplastic brain lesions. The results also under-score the diversity among reactive astrocytes expressing glial fibrillary acidic protein dur-ing gliosis [109]. Previous studies have demonstrated that PDPN, a sialoglycoprotein found in the cell membrane and encoded by the PDPN gene, is upregulated and linked to cellular invasion in astrocytic tumors, although the regulatory mechanisms remain un-known [109].
- Lines # 547-557: Cancers that have PDPN expression tend to be more deadly, as they have an increased capacity to generate stem cells, invade surrounding tissues, and transition from epithelial to mesenchymal cells. This transition contributes to the aggressive advancement of the cancer [104]. In a 2015 study by Grau et al., two human glioma cell lines (U373MG and U87MG) were genetically altered to express PDPN. The effectiveness of this alteration was confirmed through FACS analysis, PCR, and immunocytochemistry. It was observed that the level of PDPN expression varied depending on the grade of the tumors, and all glio-blastomas tested positive for PDPN. Moreover, the study indicated that PDPN expression resulted in increased tube formation activity in endothelial cells, regardless of VEGF. These findings suggest that PDPN expression may have a significant impact on tumor progression [105].
- Lines # 559-576: Putthisen et al. (2022), identified U373-GSC, a stem-like cell associated with gliomas (GSC). This GSC expressed a specific type of glycan modification involving sialic acid and was common in both GSCs and the parent cell lines [106]. The study demonstrated that the sialic acid modification was highly expressed in GSCs and significantly reduced during the differentiation of GSCs into glioma cells or in the parental cells. [106]. This study con-firms that sialic acid modifications (MAL-SGs/alpha-2,3 sialylations) play a role in the survival and maintenance of GSCs and inhibiting these modifications could lead to the apoptosis of GSCs [106]. In a study, an anti-PDPN monoclonal antibody, humLpMab-23, was humanized, and a defucosylated form, humLpMab-23-f, was produced to enhance its effectiveness in targeting PDPN-overexpressed cells, particularly in GBM. The study demonstrated that humLpMab-23-f was able to induce cellular cytotoxicity and exerted high antitumor activity in mouse xenograft models, suggesting its potential as an antibody therapy for PDPN-positive glioblastomas [104]. Another study investigated the expression of PDPN in glioma tissues and its association with prognostic factors. As a result, PDPN was found to be overexpressed in the majority of glioma tissues compared to normal tis-sues and was positively correlated with certain prognostic factors such as TERT promoter mutation status, and ATRX retention status. Additionally, PDPN knockdown was found to suppress proliferation and reduce protein expression in glioma cells [107].
- Lines # 465-470: Eisemann et al (2018) deleted PDPN in patient-derived human GBM cells and found a gene signature associated with high PDPN expression in tumor cells, indicating a poor outcome [97]. Results from the study conducted by Wang et al (2021) were similar to Kapteijn et al. (2023) i.e., high PDPN expression triggers platelet activation and is associated with an increased risk of VTE in patients with cancer [94,98]. Such PDPN-associated mechanisms underlying VTE in patients with GBM highlight the potential implications for cancer-associated venous thrombosis.
- A more compact and incisive presentation of the literature would greatly improve the present version of the review.
We have re-written entire manuscript and removed any duplication
Response:
- Minor comment- 22 row 625 " The glycoprotein PDPN (PDPN)". the abbreviation PDPN was defined long before you can use it without further indications.
Response: Yes, we have used abbreviation PDPN in the entire manuscript (wherever applicable)

Reviewer 2 Report
Comments and Suggestions for Authors
In this manuscript, Sharma et al., reviewed the current research of podoplanin (PDPN) in advancing the progression and metastasis of glioblastoma multiforme (GBM). While the topic is interesting, there are several aspects that can be further improved. I listed my comments below.
1. In the ‘Simple Summary’ section, they wrote ‘there is a lack of strong clinical evidence to show PDPN as an independent prognostic marker in GBM. Therefore, we have decided to present this review’. Here it is hard to understand the causal relationship of the two sentences linked by ‘therefore’.
2. In ‘Abstract’. L22 – 23, the authors wrote ‘Based on recent research, its expression has been associated with poor prognosis in GBM’. This seems contradictory to what they wrote in the ‘Simple Summary’.
3. L44, the 173,690 in males and 148,032 in females add up to 321,722, instead of 321,731 as the authors wrote. What are the remaining 9 cases?
4. Figure 1, the use of pie chart to illustrate this is not correct. The green (both sexes) already include males and females, therefore further adding them together makes no sense.
5. Table 2 and 3 summaries the clinical studies on GBM and PDPN, respectively. The authors state in L525 – 527, that they didn’t find any study of PDPN in GBM. In this case, listing all those clinical studies in detail seems irrelevant to the paper’s topic.
6. Reading the whole manuscript, I have a feeling that the first half focuses on GBM and the second half on PDPN. In fact, based on current studies, the role of PDPN in GBM remains poorly understood. I suggest that the authors change the title of the manuscript and focus on discussing GBM and PDPN separately (as they did in the main text of the manuscript), and then provide some insightful perspective on the potential role of PDPN in GBM. This would make things look smoother and more logical.
Comments on the Quality of English Language
More scientific writing can be applied to the figure and table legends.
Author Response
In this manuscript, Sharma et al., reviewed the current research of podoplanin (PDPN) in advancing the progression and metastasis of glioblastoma multiforme (GBM). While the topic is interesting, there are several aspects that can be further improved. I listed my comments below.
- In the ‘Simple Summary’ section, they wrote ‘there is a lack of strong clinical evidence to show PDPN as an independent prognostic marker in GBM. Therefore, we have decided to present this review’. Here it is hard to understand the causal relationship of the two sentences linked by ‘therefore’.
Thanks for your comments:
Correct statement (highlighted in green): We don't have enough review literature combining pre-clinical and clinical evidence to demonstrate PDPN as a standalone prognostic indicator in GBM. This is why we have chosen to present this review.
- In ‘Abstract’. L22 – 23, the authors wrote ‘Based on recent research, its expression has been associated with poor prognosis in GBM’. This seems contradictory to what they wrote in the ‘Simple Summary’.
Correct statement (highlighted in green): In recent research studies, its expression has been linked with prognosis in GBM.
- L44, the 173,690 in males and 148,032 in females add up to 321,722, instead of 321,731 as the authors wrote. What are the remaining 9 cases?
Correct statement (highlighted in green): In 2022, the global incidence of brain and other CNS-associated cancers in both genders was 321,731 (173,690 males and 148,032 females).
- Figure 1, the use of pie chart to illustrate this is not correct. The green (both sexes) already include males and females, therefore further adding them together makes no sense.
The purpose of pie chart is to simply show that the occurrence of GBM in males is higher than in females.
- Table 2 and 3 summarize the clinical studies on GBM and PDPN, respectively. The authors state in L525 – 527, that they didn’t find any study of PDPN in GBM. In this case, listing all those clinical studies in detail seems irrelevant to the paper’s topic.
One of our objectives is to encourage researchers by showing that clinical evidence for PDPN and GBM are available but not together. Hence, further exploration is required.
- Reading the whole manuscript, I have a feeling that the first half focuses on GBM and the second half on PDPN. In fact, based on current studies, the role of PDPN in GBM remains poorly understood. I suggest that the authors change the title of the manuscript and focus on discussing GBM and PDPN separately (as they did in the main text of the manuscript), and then provide some insightful perspective on the potential role of PDPN in GBM. This would make things look smoother and more logical. We have revised and organized the entire manuscript including Title.
Round 2
Reviewer 2 Report
Comments and Suggestions for Authors
The authors did an efficient job, completing a major revision in 4 days. However, there are remaining concerns the authors didn’t address.
1. The authors didn’t make any changes to Fig. 1. As I mentioned, the pie chart makes no sense.
2. The authors didn’t provide with the explanation for L44, the 173,690 in males and 148,032 in females add up to 321,722, instead of 321,731 as the authors wrote. What are the remaining 9 cases?
Author Response
Greetings,
Thanks for providing your comments:
- The authors didn’t make any changes to Fig. 1. As I mentioned, the pie chart makes no sense.
Response: We have removed figure 1 with Pie chart
- The authors didn’t provide with the explanation for L44, the 173,690 in males and 148,032 in females add up to 321,722, instead of 321,731 as the authors wrote. What are the remaining 9 cases?
Response: We have re-written entire manuscript. Please see the changes below:
- Please see the L44 of modified version: In 2022, the global incidence of brain and other CNS-associated cancers was 173,690 in males and 148,032 in females [3].
- We have removed the sentence or information on 9 cases. Current version doesn’t have any such information.
- Lines # 37-60: Glioblastoma multiforme (GBM) is a frequently occurring malignant primary brain tumor that involves diffusive invasion in the surrounding brain tissues [1]. It is the most aggressive type of brain tumor, categorized as grade 4 astrocytic glioma by the World Health Organization (WHO). It typically affects people between 55 and 84 years old and has a high mortality rate [1]. In 2024, it is projected that there will be 2,001,140 new cases of cancer and 611,720 cancer-related deaths in the United States [2]. In 2023, there were 1,958,310 new cancer cases and 609,820 projected cancer-related deaths in the United States [2]. In 2022, the global incidence of brain and other CNS-associated cancers was 173,690 in males and 148,032 in females [3]. Data from 2014 to 2020 in the United States showed that the prevalence of brain and other nervous system-related cancers varied with age in both males and females across all races and ethnicities. The highest prevalence rate was observed in the 0-19 age group as shown in Figure 1 [4]. According to the Internation-al Agency for Research on Cancer (IARC) in 2022, the highest mortality associated with brain and other CNS cancers in both genders was found in Asia, followed by Europe, Lat-in America, the Caribbean, North America, Africa, and Oceania [5], as shown in Figure 2. From 2016 to 2020, the average annual incidence rate of all brain and central nervous sys-tem (CNS) tumors was 24.83 per 100,000 people, with higher rates in females than males and non-Hispanic than Hispanic population [6]. GBM is more common in males, while meningioma is more common in females [7]. Fifty percent of adult GBM patients die within 10–12 months after diagnosis, and only about 10% survive 24 months after diagnosis [8]. As survival time increases, patients with GBM may experience cognitive issues, neurolog-ical deficits due to radiation-induced tissue damage, communicating hydrocephalus, and, in some cases, cranial neuropathies and polyradiculopathies caused by the spread of cancer cells to the meninges covering the brain and spinal cord [9,10].

Round 3
Reviewer 2 Report
Comments and Suggestions for Authors
No further comments.
Author Response
Greetings,
As mentioned by the reviewer: No further comments.
Thanks!,